# The Psychological Impact of COVID-19 among a Sample of Italian Adults with High-Functioning Autism Spectrum Disorder: A Follow-Up Study

**DOI:** 10.3390/healthcare10050782

**Published:** 2022-04-22

**Authors:** Veronica Nisticò, Giulia Fior, Raffaella Faggioli, Alberto Priori, Orsola Gambini, Benedetta Demartini

**Affiliations:** 1Dipartimento di Scienze della Salute, Università degli Studi di Milano, 20142 Milano, Italy; veronica.nistico@unimi.it (V.N.); alberto.priori@unimi.it (A.P.); orsola.gambini@unimi.it (O.G.); 2“Aldo Ravelli” Research Center for Neurotechnology and Experimental Brain Therapeutics, Università degli Studi di Milano, 20142 Milano, Italy; 3Dipartimento di Psicologia, Università degli Studi di Milano-Bicocca, 20126 Milano, Italy; 4Unità di Psichiatria 52, ASST Santi Paolo e Carlo, Presidio San Paolo, 20142 Milano, Italy; giulia.fior@asst-santipaolocarlo.it (G.F.); rafbeans@gmail.com (R.F.); 5III Clinica Neurologica, ASST Santi Paolo e Carlo, Presidio San Paolo, 20142 Milano, Italy

**Keywords:** COVID-19, SARS-CoV-2, lockdown, Autism Spectrum Disorder, stress, anxiety, depression, PTSD

## Abstract

The strict lockdowns imposed to contain the COVID-19 pandemic brought an increase in levels of stress, anxiety, and depression in the general population. However, in a previous study, our group found that individuals with High-Functioning Autism Spectrum Disorders (HF-ASD) reported an increase in their psychological wellbeing and a decrease in their daily tiredness, in relation to the social distancing measures imposed during the first Italian lockdown (between March and May 2020). In this follow-up study, conducted during the “second wave” of COVID-19, we included the same group of individuals with HF-ASD and evaluated their levels of stress, anxiety, depression, PTSD-related symptoms, tiredness, and perceived wellbeing; moreover, we compared our results to the ones we obtained during the first lockdown on the same population. We found that individuals with HF-ASD experienced higher levels of the aforementioned psychiatric symptoms during the second lockdown, with respect to the first one. These levels positively correlated with their scores at the Autism Quotient subscale Attention Switching: hence, we speculated that these symptoms might be due not only to the prolonging of the social distancing measures, but also to the uncertainty that HF-ASD participants started experiencing at the end of the first lockdown.

## 1. Introduction

Autism Spectrum Disorder (ASD) is a common neuropsychiatric condition characterized by persistent deficits in social interaction, communication difficulties, and specific or repetitive patterns of behaviors or interest [1]. It can be thought as a continuum, ranging from a pole with severe developmental disabilities to a pole without intellectual disability (Intelligence Quotient > 70), previously defined as High-Functioning ASD (HF-ASD) [2]. Along with a deficit in understanding and regulating their own emotions, subjects with HF-ASD also present specific social difficulties, such as interpreting what other people are thinking or feeling, interpreting others’ facial expressions, body language, or social cues and maintaining the natural give-and-take of a conversation [3]. These difficulties clearly have a negative impact on the global functioning of individuals with HF-ASD, especially in areas related to work activity and sociality where interpersonal interactions are required [4], ultimately leading them to experience severe symptoms of anxiety when they are obliged to face social situations [5]. As a matter of fact, research studies showed that individuals with HF-ASD perform better in environments where they can work alone with a high degree of autonomy in a defined and intellectually challenging job [6]. Due to the COVID-19 outbreak, at the beginning of 2020, governments worldwide were forced to impose a strict lockdown to try to contain the spread of the pandemic, which implied a significant decrease in social interactions. On one hand, it was widely demonstrated that the pandemic and the consequent containment measures have caused severe damages to the population worldwide under physical, social, and psychological perspectives: levels of stress, anxiety, and depression have greatly increased since the beginning of the outbreak, especially in elderly people, healthcare workers, and individuals with preexistent psychiatric conditions [7,8,9,10,11]. On the other hand, in a previous study, we found that individuals with HF-ASD, compared to a group of neurotypical adults, reported feeling subjectively more comfortable during the lockdown period than before and arrived at the end of their study/work day significantly less tired during the lockdown than during the month before. We speculated that subjects with HF-ASD might have somehow benefitted from the lockdown, in terms of feeling less exposed to everyday social interaction and hence feeling more comfortable and less tired. Moreover, we found that, although their levels of anxiety and depression were significantly higher than the control group, these levels were not increased compared to results of studies conducted in pre-epidemic times [12]. It remained unclear whether these results were only an acute response to the first lockdown, or a reflection of a stable trait. This also suggests that individuals with HF-ASD might have a better quality of life if specific measures involving being less exposed to everyday social interaction (such as a permanent teleworking condition) would be implemented. 

From November 2020 to the end of April 2021, to face the so-called “second wave” of the COVID-19 pandemic, the Italian Government imposed a complex series of social distancing measures potentially changing every two weeks (in terms of being more or less strict) according to the weekly trend of new infections (see Figure 1 for further explanation). 

### Aims of the Study

The aim of the present follow-up study was to evaluate the levels of stress, anxiety, depression, tiredness, and perceived well-being in a population of individuals with HF-ASD during the second Italian lockdown (t2), and to compare them to the results we obtained during the first lockdown on the same population (t1). For this reason, the same participants of t1 were contacted again and were asked to undergo the same series of self-report questionnaires they completed at t1. Three strongly validated scales were chosen to investigate feelings of depression, stress, anxiety (the Depression, Anxiety, and Stress Scale—21 items—and the Perceived Stress Scale), and symptoms suggestive of Post-Traumatic Stress Disorder (the Impact of Event Scale—Revised). Moreover, a series of ad hoc questions were asked, to investigate the subjective feeling of tiredness and psychological wellbeing of each individual. 

## 2. Materials and Methods

### 2.1. Participants

Participants with HF-ASD were recruited from the tertiary level outpatient clinic of ASST Santi Paolo e Carlo, Presidio San Paolo in Milan. They were previously diagnosed with HF-ASD by a psychiatrist and a psychologist according to DSM-5 criteria [1]. To further confirm the diagnosis, all subjects underwent: (i) the Module 4 of the Autism Diagnostic Observation Schedule—2nd version (ADOS-2) [13], a reliable semi-structured diagnostic tool based on interviewer’s clinical observation; specifically, Module 4 is validated for adults and adolescents with fluent language; (ii) the Autism Quotient (AQ) [14], a 50-item self-report questionnaire measuring the degree to which an adult without intellectual disabilities presents autistic traits; (iii) the Ritvo Autism Asperger Diagnostic Scale—Revised (RAADS-R) [15], an 80-item validated instrument designed to assist clinicians diagnosing ASD in adults.

At t1, 45 individuals with HF-ASD took part in our study. At t2, all participants were re-contacted by means of phone call or email by their clinicians; 44 of them responded and agreed to take part in the follow-up study, while one participant never responded and was considered as drop-out. As at t1, exclusion criteria were: (i) age less than 18 years old; (ii) presence of intellectual disabilities (Intelligence Quotient < 70), measured via the Wechsler Adult Intelligence Scale—Fourth Edition (WAIS-IV) [16]; (iii) psychotic disorders. Data collection took place between February and March 2021 (i.e., towards the end of the second Italian lockdown). All participants signed an online-written informed consent form before completing the questionnaire and were free to withdraw from the study at any time without giving further explanation. The study was approved by the ASST Santi Paolo e Carlo Ethics Committee (“Comitato Etico Milano Area 1”).

### 2.2. Psychometric Assessment and Ad-Hoc Questionnaire

Participants completed the Italian version of the following self-report questionnaires. First, the Depression, Anxiety and Stress Scale—21 items (DASS-21), a validated questionnaire assessing depressive and anxiety symptoms [17]. A Total Score was calculated as an index of general distress, together with three subscales: Stress, Anxiety, and Depression. According to each subscale score, participants were labelled on a severity scale; in particular, the subscale Stress score was divided into 0–7 (normal), 8–9 (mild), 10–12 (moderate), 13–16 (severe), and ≥17 (extremely severe); the subscale Anxiety score was divided into 0–3 (normal), 4–5 (mild), 6–7 (moderate), 8–9 (severe), and ≥10 (extremely severe); the subscale Depression score was divided into 0–4 (normal), 5–6 (mild), 7–10 (moderate), 11–13 (severe), and ≥ 14 (extremely severe). Second, the Impact of Event Scale—Revised (IES-R), a 22-item self-report scale that assesses subjective distress caused by traumatic events [18]. The IES-R Total Score, obtained by summing the answers to each item, was divided into 0–23 (normal), 24–32 (mild psychological impact), 33–36 (moderate psychological impact), and >37 (severe psychological impact). Moreover, three subscales were calculated, providing an indication of the level of distress experienced: Intrusion, Avoidance and Hyperarousal. Third, the Perceived Stress Scale (PSS), a 10-item validated instrument measuring “the degree to which situations in one’s life are appraised as stressful” [19]; each item is rated on a 5-point Likert scale, ranging from 0 (never) to 4 (very often). The PSS Total Score ranges from 0 to 40, with higher scores indicating higher levels of perceived stress, and it was divided into 0–13 (low stress), 14–26 (moderate stress), and 27–40 (high perceived stress). Fourth, an ad hoc questionnaire designed for the study, including three questions to be answered on a 7-point Likert scale; three distinct variables where extrapolated out of each question, as follows: (i) Tiredness Summer 2020: “How tired did you feel when you arrived at the end of your study/work day between May and November 2020, on a scale from 1 to 7 (where 1 = not tired at all and 7 = extremely tired)?”; (ii) Tiredness 2nd lockdown: “How tired have you felt at the end of your study/work day since November 2020, on a scale from 1 to 7 (where 1 = not tired at all and 7 = extremely tired)?”; (iii) Psychological Wellbeing 2nd lockdown: “How your psychological well-being has been influenced by the prolonging of the measures ordered by Italian authorities to contain the COVID-19 outbreak, which implied a clear decrease in social interactions, on a scale from 1 to 7 (where 1 = I feel less comfortable than before and 7 = I feel more comfortable than before)?”

### 2.3. Statistical Analysis

Statistical analysis was performed using SPSS version 27 (Statistical Package for Social Science). Significance level was set at α = 0.05, and all tests were 2-tailed. First, descriptive statistics were calculated for sociodemographic information and for the psychometric assessment. Second, a *t*-test for paired samples was run to compare the results of the psychometric assessment at t1 and t2 (for the complete t1 results, see [12]). Third, levels of tiredness were analyzed via Repeated Measure ANOVA, with “Tiredness” as four-level within-subject variable: (i) Tiredness pre-lockdown (collected at t1), (ii) Tiredness 1st lockdown (collected at t1), Tiredness Summer 2020 (collected at t2), and (iv) Tiredness 2nd lockdown (collected at t2). Finally, Pearson’s correlational analyses were run between the diagnostic questionnaires (AQ and RAADS-R) and the DASS-21, IES-R, PSS and ad hoc questionnaires, to assess whether the psychiatric symptoms shown during the second lockdown were associated with autistic traits.

## 3. Results

### 3.1. Sociodemographic Information

Mean age of the participants at t2 was 38.3 (S.D. 13.3, range: [20–60 years old]). In total, 28 participants were male, 15 were female, 1 was non-binary. All participants were of Caucasian ethnicity. Since November 2020 to the days of data collection, two participants were infected with COVID-19 (of whom one remained asymptomatic, one showed symptoms). A total of 13 participants declared that some of their family members were infected with COVID-19 (of whom 3 co-habited with the participants, 10 did not). Three participants had already received the first dose of the COVID-19 vaccine. With respect to their working activity, 11 participants were physically going to work, 23 worked or studied from home, and 10 were unemployed. At the time of testing, 22 participants were treated with psychotherapy on video conference. Further details are reported in Table 1 and Table 2.

### 3.2. Psychometric Assessment

During the second lockdown (t2), at the DASS-21, 32 participants (72.7%) showed mild-to-extremely severe levels of stress, 23 participants (52.3%) mild-to-extremely severe levels of anxiety, 34 participants (77.3%) mild-to-extremely severe levels of depression; at the IES-R, 22 participants (50%) showed a severe psychological impact and PTSD-like symptomatology, and at the PSS 39 participants (88.6%) reported moderate-to-high levels of stress.

Comparing t2 and t1 data, it emerged that at the DASS-21, participants showed significantly higher scores at t2 than at t1, at the Total Score (t = - 3.276, df = 43, *p* = 0.002), and all the subscales: Stress (t = −3.705, df = 43, *p* = 0.001), Anxiety (t = −2.581, df = 43, *p* = 0.013), Depression (t = −2.333, df = 43, *p* = 0.024). At the IES-R, participants showed significantly higher scores at t2 than at t1 at the Total Score (t = −2.433, df = 43, *p* = 0.019) and at the subscale Intrusion (t = −2.736 df = 43, *p* = 0.009), but not at the subscales of Avoidance (t = −1.624, df = 43, *p* = 0.112) and Hyperarousal (t = −1.877, df = 43, *p* = 0.067 (trend)). At the PSS, only a trend towards significance emerged (t = −1.767, df = 43, *p* = 0.084), with the scores at t2 being higher than t1. Further details are reported in Table 3.

### 3.3. Ad-Hoc COVID-19 Questionnaire

RM ANOVA showed that there was a significant effect of the within-subject factor *Tiredness* (F (3, 126) = 6.957, *p* < 0.001, eta^2^ = 0.976). In particular, a significant quadratic trend emerged (F (1, 42) = 20.335, *p* < 0.001): on a scale from 1 (not tired at all) to 7 (extremely tired), participants scored with an average of 5.16 (S.D. = 1.599); this score significantly got lower during the first lockdown (average = 4.02, S.D. = 2.04), and gradually increased again during Summer 2020 (average = 4.26, S.D. = 1.399) and the second lockdown (average 4.84, S.D. = 1.703) (Figure 2). Finally, levels of the participants’ reported psychological wellbeing were significantly lower at t2, with respect to t1 (t = 3.090, df = 42, *p* = 0.004), (Table 3).

### 3.4. Correlational Analyses

The AQ Total Score positively correlated with the PSS (r = 0.437, *p* = 0.005) and with the Tiredness Summer 2020 (r = 0.321, *p* = 0.049). Similarly, the AQ subscale Social Skills positively correlated with the PSS (r = 0.423, *p* = 0.007) and with the Tiredness Summer 2020 (r = 0.371, *p* = 0.022). The AQ subscale Attention Switching positively correlated with most of the variable assessed: the DASS-21 Total Score (r = 0.504, *p* = 0.001) and its subscales Stress (r = 0.452, *p* = 0.004) and Depression (r = 0.520, *p* = 0.001); the IES-R Total Score (r = 0.476, *p* = 0.002) and its subscales Avoidance (r = 0.444, *p* = 0.005), Intrusion (r = 0.393, *p* = 0.013), and Hyperarousal (*p* = 0.484, *p* = 0.002); the PSS (*p* = 0.596, *p* < 0.001); the Tiredness 2nd lockdown (*p* = 0.501, *p* = 0.001) but not the Tiredness Summer 2020 (r = 0.281, *p* = 0.087). Several significant correlations also emerged between the scores at the RAADS-R and the ones at our psychometric assessment, which are fully reported in Table 4.

## 4. Discussion

The first aim of this study was to assess the levels of specific psychiatric symptoms (i.e., stress, anxiety, depression), tiredness, and perceived well-being in a group of adult individuals with HF-ASD during the second lockdown (t2), which was imposed by the Italian government between November 2020 and April 2021 to try to contain the “second wave” of the COVID-19 pandemic. We found that the majority of our participants showed mild-to-extremely severe levels of stress, anxiety, and depression according to the DASS-21 and to the PSS, and that half of our participants reported a severe psychological impact and PTSD-like symptomatology, as per the IES-R. 

Second, we aimed to compare our results to the ones we obtained with the same protocol, on the same population during the first lockdown (February–May 2020, t1), which are fully reported in detail in [12]. We found that the levels of stress, depression and anxiety, evaluated with the DASS-21, and the symptoms suggestive of PTSD evaluated with the IES-R, Intrusion in particular, were significantly higher during the second lockdown, with respect to the first. In line with these results, the levels of psychological well-being self-reported by our group of patients with HF-ASD was significantly lower than the first lockdown, and their level of tiredness at the end of their study/work day gradually increased during the Summer 2020 to almost reach, during the second lockdown, the high pre-pandemic level (Figure 1). Our results seem to suggest that, although at the very beginning of the pandemic our group of individuals with HF-ASD seemed to react better than the NA group to the social distancing measures, the prolonging of the same measures led even HF-ASD participants to experience enhanced stress, anxiety, depression, and PTSD-like symptoms. A recent study seems to corroborate our results: Maljaars and colleagues [20] investigated the experiences of adults with ASD and neurotypical adults in Belgium, the Netherlands and the United Kingdom during the first lockdown (Spring 2020) and during the following months, when the social distancing measures were alleviated (Summer 2020). They found that more autistic adults (56%) than neurotypical adults (31%) reported experiencing significantly more stress during Summer 2020 than during the first lockdown. This group of adults with ASD explained their perceived stress with reasons relating to risks due to relaxation of measures, increasing lack of clarity about the measures, constant changes in measures, a less quiet life, and an uncertain future. Maljaars and colleagues’ results are in line not only with the fact that the perceived stress and tiredness of our sample started increasing during Summer 2020, with respect to the first lockdown, but also to the fact that our results positively correlated with the AQ subscale Attention Switching. This subscale evaluates the difficulties experimented in changing the focus of one’s own attention and in flexibly modifying one’s own routine; it includes items such as the following: “It does not upset me if my daily routine is disturbed” (to be reversed); “I like to carefully plan any activities I participate in”; “New situations make me anxious” [14]. Hence, we might speculate that the symptomatology reported by our participants with HF-ASD is not simply due to the prolonging of the lockdown, but to the uncertainty that they started experiencing when the first lockdown ended: first, during Summer 2020, the alleviation of the lockdown inevitably exposed individuals to a higher risk of contagion, although significantly smaller than during the first and second wave; second, the second lockdown in Italy was characterized by a series of measures that could potentially change every two weeks, in terms of being more or less strict, according to the weekly levels of new infections registered in the country. Although necessary, this policy might have been particularly difficult to adapt to for individuals with HF-ASD. In fact, “rigidity, insistence on sameness and inflexible adherence to routines”, leading to “extreme distress at small changes and difficulties with transitions” are amongst the DSM-5 criteria for ASD diagnosis [1]. These features, amongst the others, cause significant discomfort and/or impairment, and cause individuals with ASD to need specific support in their daily activities [1]. As a matter of fact, recent evidence highlighted how the frequent changing in everyday routine was a cause of significant distress not only amongst ASD individuals [21], but also for their caregivers, who should be offered specific psychological support [22].

### Limitations

We acknowledge the limitations of our study. First, all data were self-reported. Second, we did not consider the influence of psychiatric comorbidities, such as anxiety or mood disorders, and of psychiatric medications, which might have been adjusted between t1 and at t2 and might have influenced our results. Finally, although we asked our group of participants to report whether they or their family members were infected with COVID-19 since November 2020 (i.e., the beginning of the “second wave”), we did not ask them to provide the same data for the period between May and November 2020.

## 5. Conclusions

In conclusion, in our study, we found that individuals with HF-ASD experienced higher levels of stress, anxiety, depression, PTSD-related symptoms, and tiredness during the second COVID-19 lockdown, which occurred between November 2020 and April 2021, with respect to the first one (March–May 2020). These levels of symptomatology positively correlated with the scores at the AQ subscale Attention Switching, which led us to speculate that the symptomatology reported by our participants with HF-ASD is not simply due to the prolonging of the social distancing measures, but to the uncertainty that individual’s with HF-ASD started experiencing at the end of the first lockdown.

## Figures and Tables

**Figure 1 healthcare-10-00782-f001:**
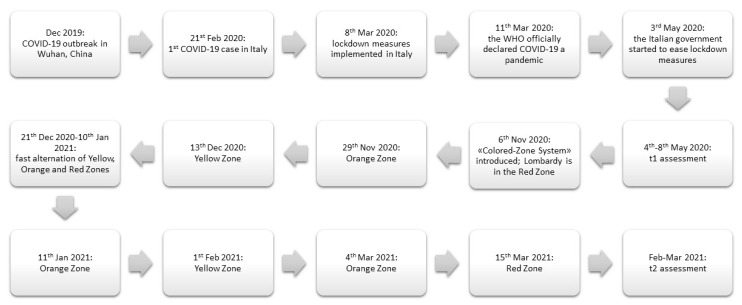
Timeline of the evolution of the COVID-19 pandemic and of the consequent restrictions implemented in Italy during the first and second wave. Since the beginning of the second wave (November 2020), the Italian government implemented a so-called “Colored-Zone System”: every two weeks, the spread of the COVID-19 contagion was evaluated separately in each region of Italy and, according to its severity, different levels of restrictive measures were imposed. In particular, the “Red Zone” corresponded to a complete lockdown; the “Orange Zone” corresponded to mild-to-severe restrictive measures (e.g., it was possible for citizens to move freely only within their town of residency and only between 5 am and 10 pm); the “Yellow Zone” included mild restrictive measures (e.g., everyone could move freely within their own region, but still had to respect the 10 pm–5 am curfew). Wearing a face mask always remained mandatory.

**Figure 2 healthcare-10-00782-f002:**
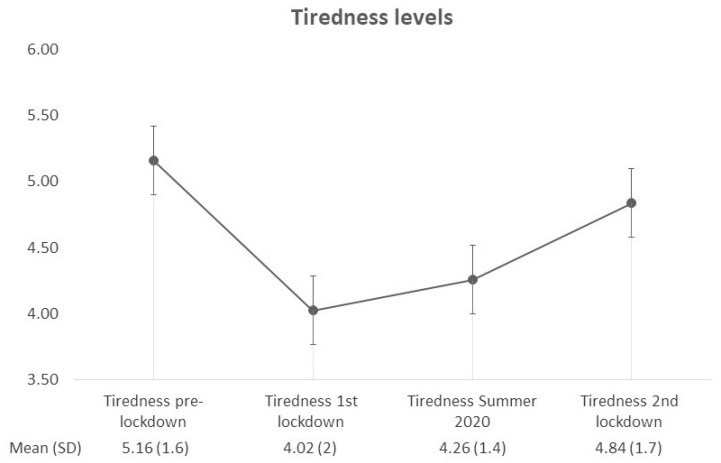
Tiredness levels of our sample of individuals with HF-ASD at the four time-points assessed. Abbreviations: SD = Standard Deviation.

**Table 1 healthcare-10-00782-t001:** Demographic features of our sample of individuals with HF-ASD.

		Value
Age, Mean (SD)		38.3 (12.3)
Gender, N (%)	M	28 (63.6)
	F	15 (34.1)
	Non binary	1 (2.3)
ADOS-2 Communication, mean (SD)		4.27 (1.98)
ADOS-2 Reciprocal social interaction, mean (SD)		7.52 (2.61)
ADOS-2 Imagination/Creativity, mean (SD)		1.39 (0.69)
ADOS-2 Stereotyped behaviors and restricted interests, mean (SD)		1.39 (1.28)
ADOS-2 Total Social Communication, mean (SD)		11.8 (4.21)
AQ Total Score, mean (SD)		33.05 (7.63)
AQ Social skills, mean (SD)		6.9 (2.51)
AQ Attention switching, mean (SD)		7.97 (1.87)
AQ Attention to detail, mean (SD)		6.64 (2.15)
AQ Communication, mean (SD)		6.23 (2.28)
AQ Imagination, mean (SD)		5.31 (2.12)
RAADS-R Total Score, mean (SD)		136.56 (36.92)
RAADS-R Social Relatedness, mean (SD)		66.41 (19.36)
RAADS-R Circumscribed Interests, mean (SD)		27.23 (7.77)
RAADS-R—Language, mean (SD)		10.44 (4.17)
RAADS-R Sensory-motor, mean (SD)		32.49 (12.17)

Abbreviations: ADOS-2 = Autism Diagnostic Observation Schedule—2nd version; AQ = Autism Quotient; RAADS-R = the Ritvo Autism Asperger Diagnostic Scale—Revised; SD = Standard Deviation.

**Table 2 healthcare-10-00782-t002:** Demographic features of our sample of individuals with HF-ASD.

Activity during lockdown, N (%)	Working mostly away from home	11 (25)
	Working mostly from home	23 (52.3)
	Unoccupied	10 (22.7)
Received COVID-19 Vaccination, N (%)	Yes	3 (6.8)
	No	41 (93.2)
DASS-21 Stress, N (%)	Normal	12 (27.3)
	Mild	4 (9.1)
	Moderate	9 (20.5)
	Severe	12 (27.3)
	Extremely severe	7 (15.9)
DASS-21 Anxiety, N (%)	Normal	21 (47.7)
	Mild	7 (15.9)
	Moderate	4 (9.1)
	Severe	3 (6.8)
	Extremely severe	9 (20.5)
DASS-21 Depression, N (%)	Normal	10 (22.7)
	Mild	3 (6.8)
	Moderate	8 (18.2)
	Severe	7 (15.9)
	Extremely severe	16 (36.4)
IES-R Total Score, N (%)	Normal	18 (40.9)
	Mild psychological impact	3 (6.9)
	Moderate psychological impact	1 (2.3)
	Severe psychological impact	22 (50)
PSS, N (%)	Low stress	5 (11.4)
	Moderate stress	21 (47.7)
	High stress	18 (40.9)

Abbreviations: DASS-21 = Depression, Anxiety and Stress Scale—21 items; IES-R = Impact of Event Scale—Revised; PSS = Perceived Stress Scale; SD = Standard Deviation.

**Table 3 healthcare-10-00782-t003:** Psychometric assessment at t1 and t2.

	t1	t2	*p*
DASS-21 Total Score, mean (SD)	19.48 (10.03)	26.5 (14.57)	0.002
DASS-21 Stress, mean (SD)	7.68 (5.55)	10.91 (5.37)	0.001
DASS-21 Anxiety, mean (SD)	3.48 (4.21)	5.16 (4.8)	0.031
DASS-21 Depression, mean (SD)	8.32 (6.19)	10.43 (6.54)	0.024
IES-R Total Score, mean (SD)	25.75 (16.55)	31.61 (20.66)	0.019
IES-R Avoidance, mean (SD)	1.09 (0.71)	1.27 (0.9)	0.112
IES-R Intrusion, mean (SD)	1.13 (0.93)	1.49 (1.06)	0.009
IES-R Hyperarousal, mean (SD)	1.33 (0.86)	1.59 (1.1)	0.067
PSS, mean (SD)	22.48 (8.14)	24.18 (8.03)	0.084
Psychological well-being	4.67 (1.91)	3.72 (1.533)	0.004

Abbreviations: DASS-21 = Depression, Anxiety and Stress Scale—21 items; IES-R = Impact of Event Scale—Revised: PSS = Perceived Stress Scale: SD = Standard Deviation.

**Table 4 healthcare-10-00782-t004:** Correlational analysis.

		DASS-21 Stress	DASS-21 Anxiety	DASS-21 Depression	DASS-21 Total Score	IES-R Avoidance	IES-R Intrusion	IES-R Hyperarousal	IES-R Total Score	PSS	Tiredness Summer 2020	Tiredness 2nd Lockdown	Psychological Wellbeing
AQ Total Score	r	0.278	0.128	0.227	0.247	0.212	0.235	0.241	0.248	0.437 *	0.321 *	0.251	0.203
	*p*	0.087	0.438	0.165	0.129	0.196	0.149	0.140	0.128	0.005	0.049	0.129	0.221
AQ Social Skills	r	0.135	0.123	0.186	0.174	0.179	0.201	0.174	0.201	0.423 *	0.371 *	0.192	0.307
	*p*	0.412	0.457	0.257	0.289	0.276	0.219	0.290	0.219	0.007	0.022	0.248	0.061
AQ Attention Switching	r	0.452 *	0.308	0.520 *	0.504 *	0.444 *	0.393 *	0.484 *	0.471 *	0.596 *	0.281	0.501 *	−0.007
	*p*	0.004	0.057	0.001	0.001	0.005	0.013	0.002	0.002	0.000	0.087	0.001	0.968
AQ Attention To Detail	r	0.133	−0.007	−0.037	0.030	0.055	−0.043	−0.059	−0.017	0.116	0.249	0.097	0.135
	*p*	0.418	0.966	0.824	0.856	0.740	0.795	0.722	0.920	0.483	0.132	0.563	0.419
AQ Communication	r	0.227	0.145	0.145	0.197	0.165	0.247	0.194	0.222	0.215	0.098	0.177	0.145
	*p*	0.165	0.380	0.379	0.230	0.315	0.130	0.237	0.175	0.189	0.558	0.289	0.384
AQ Imagination	r	0.062	−0.106	0.017	−0.004	−0.076	0.038	0.084	0.015	0.196	0.112	−0.051	0.082
	*p*	0.710	0.522	0.917	0.980	0.645	0.817	0.611	0.928	0.232	0.503	0.763	0.626
RAADS-R Total Score	r	0.358 *	0.390 *	0.359 *	0.422 *	0.344 *	0.431 *	0.438 *	0.438 *	0.536 *	0.294	0.270	0.213
	*p*	0.025	0.014	0.025	0.007	0.032	0.006	0.005	0.005	0.000	0.073	0.101	0.199
RAADS-R Social Relatedness	r	0.246	0.280	0.284	0.311	0.318 *	0.367 *	0.297	.358 *	0.464 *	0.242	0.089	0.119
	*p*	0.130	0.084	0.080	0.054	0.049	0.022	0.066	0.025	0.003	0.143	0.596	0.476
RAADS-R Circumscribed Interests	r	0.375 *	0.345 *	0.283	0.380 *	0.276	0.338 *	0.412 *	0.367 *	0.451 *	0.302	0.371 *	0.221
	*p*	0.019	0.031	0.081	0.017	0.089	0.035	0.009	0.022	0.004	0.065	0.022	0.182
RAADS-R Language	r	0.314	0.353 *	0.147	0.298	0.175	0.318 *	0.357 *	0.305	0.392 *	−0.106	−0.025	0.137
	*p*	0.052	0.027	0.372	0.066	0.285	0.049	0.026	0.059	0.014	0.525	0.882	0.412
RAADS-R Sensory-motor	r	0.348 *	0.396 *	0.405 *	0.442 *	0.302	0.399 *	0.471 *	0.420 *	0.465 *	0.349 *	0.446 *	0.268
	*p*	0.030	0.013	0.010	0.005	0.061	0.012	0.002	0.008	0.003	0.032	0.005	0.104

Abbreviations: ADOS-2 = Autism Diagnostic Observation Schedule—2nd version; AQ = Autism Quotient; DASS-21 = Depression, Anxiety, and Stress Scale—21 items; IES-R = Impact of Event Scale—Revised: PSS = Perceived Stress Scale; r = Pearson’s r; RAADS-R = the Ritvo Autism Asperger Diagnostic Scale—Revised; SD = Standard Deviation; * = *p* < 0.05.

## Data Availability

The data presented in this study are available upon request by any qualified investigator.

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
