# Peer review of "The Psychological Impact of COVID-19 among a Sample of Italian Adults with High-Functioning Autism Spectrum Disorder: A Follow-Up Study"

_healthcare, 2022, doi:10.3390/healthcare10050782_

Round 1
Reviewer 1 Report
This is an interesting study in a cohort of stable patients with high functioning autism spectrum disorder.
It should be stated that all rating scales were translated and validated for use in Italian if this is the case.
Psychotic disorders were mentioned as an exclusion critrerion, suggesting that the patients may have had other psychiatric comorbidities (such as mood or anxiety disorders). Were any of the participants on psychiatric medication? If so, could their medication have been adjusted between t1 and t2? Is it possible that any of the subjects developed COVID-19 (or any other acute physical illness) during the data collection period or at some point between t1 and t2? If such data is unavailable it should be noted as a limitation of the study as an important confounding variable.
If most psychiatric comorbidities were not excluded it would be interesting to study the difference in the prevalence of mood and anxiety disorders between t1 and t2 in this cohort, as well as the incidence of new psychopathology during the lockdowns and the summer in between. If such data has already been collected it would make for an interesting study.
The observation that individuals with ASD have difficulty in adapting to changing situations is suggested as an explanation for the differences as detected in the rating scales between the end of the first and second lockdowns. This claim would be further supported if the authors provided a timeline of the introduction and repeal of various COVID containment measures in the Milan area from the first lockdown up to the period of data collection as readers not fluent in the Italian language would not be easily able to look up this data.
Author Response
We would like to thank the reviewer for the positive feedback and helpful comments which we hope have helped us to improve our manuscript. The reviewer’s comments are addressed one by one.
Comments and Suggestions for Authors
This is an interesting study in a cohort of stable patients with high functioning autism spectrum disorder.
Comment: It should be stated that all rating scales were translated and validated for use in Italian if this is the case.
Response: we did not autonomously translate the rating scales, but implemented the validated Italian versions; hence, we corrected that sentence of the Method section as follows: “Participants completed the Italian version of the following self-report questionnaires”.
Comment: Psychotic disorders were mentioned as an exclusion critrerion, suggesting that the patients may have had other psychiatric comorbidities (such as mood or anxiety disorders). Were any of the participants on psychiatric medication? If so, could their medication have been adjusted between t1 and t2? Is it possible that any of the subjects developed COVID-19 (or any other acute physical illness) during the data collection period or at some point between t1 and t2? If such data is unavailable it should be noted as a limitation of the study as an important confounding variable. If most psychiatric comorbidities were not excluded it would be interesting to study the difference in the prevalence of mood and anxiety disorders between t1 and t2 in this cohort, as well as the incidence of new psychopathology during the lockdowns and the summer in between. If such data has already been collected it would make for an interesting study.
Response: we added a limitation section and added the requested limitations, as follows: “Second, we did not consider the influence of psychiatric comorbidities, such as anxiety or mood disorders, and of psychiatric medications, which might have been adjusted between t1 and at t2 and might have influenced our results. Finally, although we asked our group of participants to report whether they or their family members were infected with COVID-19 since November 2020 (i.e., the beginning of the “second wave”), we did not ask them to provide the same data for the period between May and November 2020.”
Comment: The observation that individuals with ASD have difficulty in adapting to changing situations is suggested as an explanation for the differences as detected in the rating scales between the end of the first and second lockdowns. This claim would be further supported if the authors provided a timeline of the introduction and repeal of various COVID containment measures in the Milan area from the first lockdown up to the period of data collection as readers not fluent in the Italian language would not be easily able to look up this data.
Response: a new figure (Figure 1) was introduced, with a timeline and an explanation of the COVID-19 restrictions in Lombardy, Italy.
Reviewer 2 Report
The strict lockdowns imposed to contain the COVID-19 pandemic brought to an increase in levels of stress, anxiety, and depression in the general population.
The authors report that, in a previous study, their group found that individuals with High-Functioning Autism Spectrum Disorders (HF-ASD) reported an increase in their psychological wellbeing and a decrease in their daily tiredness, in relation to the social distancing measures imposed during the first Italian lockdown (between March and May 2020).
In this follow-up study, conducted during the “second wave” of COVID-19, the authors included the same group of individuals with HF-ASD and evaluated their levels of stress, anxiety, depression, PTSD-related symptoms, tiredness, and perceived wellbeing; moreover, we compared our results to the ones we obtained during the first lockdown on the same population.
The authors found that individuals with HF-ASD experienced higher levels of the aforementioned psychiatric symptoms during the second lockdown, with respect to the first one.
These levels positively correlated with their scores at the Autism Quotient subscale Attention Switching: hence, they speculated that these symptoms might be due not only to the prolonging of the social distancing measures, but also to the uncertainty that HF-ASD participants started experiencing at the end of the first lockdown.
The study is very interesting and well written.
A few suggestions with an academic spirit:
- Minimize the use of acronyms and insert them in a list
- Better balance the abstract with the summary of the sections.
- The aims area at the end of the introduction. I’d put them in separated section and I’d enlarge them introducing the idea of the tests etc….
- Figure 1 could be improved using a different diagram with the error bars.
- I’d separate table 1 with two tables. It is a bit hard to follow.
Congratulation to the authors for their really interesting study.
Author Response
We would like to thank the reviewer for the positive feedback and helpful comments which we hope have helped us to improve our manuscript. The reviewer's comments are addressed one by one.
Comments and Suggestions for Authors
The strict lockdowns imposed to contain the COVID-19 pandemic brought to an increase in levels of stress, anxiety, and depression in the general population.
The authors report that, in a previous study, their group found that individuals with High-Functioning Autism Spectrum Disorders (HF-ASD) reported an increase in their psychological wellbeing and a decrease in their daily tiredness, in relation to the social distancing measures imposed during the first Italian lockdown (between March and May 2020).
In this follow-up study, conducted during the “second wave” of COVID-19, the authors included the same group of individuals with HF-ASD and evaluated their levels of stress, anxiety, depression, PTSD-related symptoms, tiredness, and perceived wellbeing; moreover, we compared our results to the ones we obtained during the first lockdown on the same population.
The authors found that individuals with HF-ASD experienced higher levels of the aforementioned psychiatric symptoms during the second lockdown, with respect to the first one.
These levels positively correlated with their scores at the Autism Quotient subscale Attention Switching: hence, they speculated that these symptoms might be due not only to the prolonging of the social distancing measures, but also to the uncertainty that HF-ASD participants started experiencing at the end of the first lockdown.
The study is very interesting and well written.
A few suggestions with an academic spirit:
Comment: Minimize the use of acronyms and insert them in a list
Response: we added a list of acronyms and tried to minimize their use.
Comment: Better balance the abstract with the summary of the sections.
Response: according to the Instruction for Authors, the abstract should be unstructured. We tried to organize it in order to make it clearer to the reader.
Comment:The aims area at the end of the introduction. I’d put them in separated section and I’d enlarge them introducing the idea of the tests etc….
Response: we added a separated section with the aims, and expanded it as follows:
“1.2 Aims of the study
Aim of the present follow-up study was to evaluate the levels of stress, anxiety, depression, tiredness, and perceived well-being in a population of individuals with HF-ASD during the second Italian lockdown (t2), and to compare them to the results we obtained during the first lockdown on the same population (t1). For this reason, the same participants of t1 were contacted again and were asked to undergo the same series of self-report questionnaires they completed at t1. Three strongly validated scales were chosen to investigate feelings of depression, stress, anxiety (the Depression, Anxiety and Stress Scale – 21 items and the Perceived Stress Scale) and symptoms suggestive of Post-Traumatic Stress Disorder (the Impact of Event Scale-Revised). Moreover, a series of ad-hoc questions were asked, to investigate the subjective feeling of tiredness and psychological wellbeing of each individual.”.
Detailed description of the scales can be found in the Method section.
Comment: Figure 1 could be improved using a different diagram with the error bars.
Response: error bars were added to Figure 1 (which is now named Figure 2)
Comment:I’d separate table 1 with two tables. It is a bit hard to follow.
Response: we divided Table 1 in two tables (1 and 2); all the other tables were re-named appropriately throughout the entire manuscript.